# Cranial Shape in Infants Aged One Month Can Predict the Severity of Deformational Plagiocephaly at the Age of Six Months

**DOI:** 10.3390/jcm11071797

**Published:** 2022-03-24

**Authors:** Hiroshi Miyabayashi, Nobuhiko Nagano, Risa Kato, Takanori Noto, Shin Hashimoto, Katsuya Saito, Ichiro Morioka

**Affiliations:** 1Department of Pediatrics and Child Health, Nihon University School of Medicine, Tokyo 173-8610, Japan; miyabayashi@dr.memail.jp (H.M.); kato.risa@nihon-u.ac.jp (R.K.); noto.takanori@nihon-u.ac.jp (T.N.); morioka.ichiro@nihon-u.ac.jp (I.M.); 2Department of Pediatrics, Kasukabe Medical Center, Saitama 344-8588, Japan; fight.together.0119@gmail.com (S.H.); katsuya-saito@nifty.com (K.S.); 3Noto Children’s Clinic, Tokyo 179-0084, Japan

**Keywords:** cranial shape, deformational plagiocephaly, infant, three-dimensional scanning

## Abstract

In this study, we aimed to monitor changes in cranial shape using three-dimensional (3D) scanning to determine whether the severity of deformational plagiocephaly (DP) at the age of 6 months could be predicted at the age of 1 month. The cranial shape was measured at the ages of 1, 3, and 6 months (T1, T2, and T3, respectively) in 92 infants. We excluded those who received helmet treatment before T3. The cranial vault asymmetry index (CVAI) using 3D scanning was evaluated in all infants. DP was defined as a CVAI > 5.0% with mild (CVAI ≤ 6.25%) or moderate/severe severity (CVAI > 6.25%). The CVAI cut-off value at T1 for severe DP at T3 was determined using receiver operating characteristic (ROC) curves. At T1, T2, and T3, the respective CVAI median values were 5.0%, 5.8%, and 4.7% and the DP incidence was 50.0%, 56.8%, and 43.2%, respectively. The DP severity temporarily worsened from T1 to T2 but then improved at T3. Among the infants, 73.9% had a similar DP severity at T1 and T3 (*p* = 1.0). A ROC curve analysis revealed a CVAI cut-off value of 7.8% at T1 predicted severe DP. It was concluded that later DP severity could be predicted using 3D scanning at T1 with properly defined cut-off values.

## 1. Introduction

The human skull grows significantly during infancy. The skull of a newborn is soft and flexible because of rapid brain growth during infancy. The softness and flexibility are most prominent during the first few months of life when the sutures are still open and the head grows rapidly. During this critical period of cranial growth, infants are exposed to external factors such as orientational habits, sleeping positions, and gravity, which passively distort the cranial shape. As supine sleep is now recommended, the prevalence of deformational plagiocephaly (DP) in infants and toddlers has substantially increased [1,2]. Therefore, methods for diagnosing and measuring cranial deformities are required [3].

Cranial helmet therapy (CHT) for DP was introduced in the 2000s and has yielded good outcomes in Japan [4,5,6]. It is recommended that CHT for DP is initiated at 3–6 months of age [4,5,6,7,8,9]. A delay in starting CHT is often encountered because DP can be improved in several cases by repositioning the infant. There are no data on an appropriate timing of initiation of CHT in healthy Japanese infants; therefore, longitudinal data regarding deformity-related parameters are vital. With its introduction, many previous studies have followed the treatment course of CHT; however, only a few have studied the natural progression in healthy infants [10,11]. Furthermore, previous studies have not elucidated the progression of severe DP in patients before CHT initiation. This longitudinal study of cranial shape transformation in healthy infants is expected to provide the criteria for introducing CHT.

In this study, we aimed to longitudinally investigate the progression of skull deformities in infants aged between 1 and 6 months using a noninvasive three-dimensional (3D) scanner to determine the disease course of severe DP in infants aged 6 months, and to explore whether future DP severity could be predicted during early infancy.

## 2. Materials and Methods

This study was conducted at the Nihon University Itabashi Hospital, Kasukabe Medical Center, and Hikawadai Noto Clinic. As 3D scanners have not yet been approved as medical devices by the Pharmaceuticals and Medical Devices Agency in Japan, the ethics committee of each participating institution approved the study (Kasukabe Medical Center and Hikawadai Noto Clinic approval numbers 2019-032; Nihon University Itabashi Hospital approval number RK-200512-2). Written informed consent was also obtained from the parents of all participants when the mothers were admitted for delivery or on a follow-up visit at one month of age of the infant. 

### 2.1. Study Population

This was a prospective follow-up cohort study. Healthy infants who visited our hospitals during the study period from 1 April 2020 to 30 April 2021 were enrolled. Preterm infants (gestational age at birth < 37 weeks) and infants with neonatal asphyxia (5 min Apgar score < 7) were excluded. Asians residing in Japan were included in the study; however, those of other races were excluded.

The first scan was performed when the infant was approximately one month old (T1). Subsequently, the measurements were performed twice, at 3 and 6 months of age (T2 and T3). At T1, a questionnaire was administered to the parents to determine whether they were concerned about the skull shape of the infant. At each measurement, the pediatrician examined the skull sutures and confirmed the absence of craniosynostosis. Finally, computed tomography imaging was performed but only in cases where CHT was necessary.

### 2.2. Data Acquisition

Brief data acquisition methods have been described in our previous paper [6]. A complete 360° scan of the head including both ears was performed using a specialized 3D scanner (Artec Eva; Artec Inc., Luxembourg) whilst the head of the infant was held by the mother. The head was protected using an elastic wig cap to prevent hair tangling. The 3D scanner scanned the cranium in a continuous mode by emitting light at a maximum frequency of 16 times per second. The scanner detected the light deflected from the object, recorded the unevenness and color information of the object, and then triangulated the pattern projected onto the scanned object to create the 3D data. The 3D resolution was 0.2 mm and the accuracy was 0.1 mm. The 3D data were reconstructed by combining the overlapping regions in each of the successive scanned frames and converting them into standard triangular language files [6].

### 2.3. Data Analysis

Data were analyzed using Artec Studio image analysis software (Artec Inc., Luxembourg) and original analysis software from the Japan Medical Company (Japan Medical Company, Tokyo, Japan) to obtain 3D images and determine the cranial shape. Figure 1 shows the methods used to determine these parameters [6]. The plane connecting the sellion (SE; the lowest point of the nasal root) and the points of the left tragion (TR) and right TR (the upper margin of the tragus) was used as the reference plane (level 0). The software was used to identify the midpoint between the left and right TRs. The Y-axis was defined as the line passing through the midpoint and the SE. The X-axis was defined as the line perpendicular to the Y-axis at the midpoint of the level 0 plane (Figure 1A) [12]. The software reconstructed 10 equal cross-sections of the cranium superior to level 0. The height of each cross-section or level was determined by dividing the overall height of the infant head above the level 0 plane into 10 equal levels [4,12,13].

The level 3 plane, which has been reported to have the largest circumference [12,14], was used as the measurement plane. Figure 1B shows a cross-sectional view of level 3. The cranial asymmetry (CA) and cranial vault asymmetry index (CVAI) referred to the measurements of the differences between the longer and shorter diagonals at the level of the measurement plane at 30° from the Y-axis with regard to the length of the longer diagonal (CA = length of the longer diagonal − length of the shorter diagonal (mm); CVAI = (longer diagonal—shorter diagonal)/shorter diagonal (%)) [15]. Although the diagnostic criterion for DP is internationally defined as a CVAI > 3.5% [15,16,17], in a previous report from Japan [5]—where the prevalence of DP is higher than that of other countries—the diagnostic criterion for DP was set at a CVAI > 5% [5]. In our present study, a CVAI > 5% was also set as the diagnostic criterion for DP; the DP severity was classified as: mild, 5.00 to 6.25%; moderate, 6.25 to 8.75%; severe, 8.75–11%; and very severe, >11% [7,17].

Each cross-section was subdivided into four quadrants along the X-axis and Y-axis planes above the skull base: Q1, anterior left; Q2, anterior right; Q3, posterior right; and Q4, posterior left (Figure 1C) using image analysis software. The quadrant volumes of Q1, Q2, Q3, and Q4 comprised the sum of the volumes from levels 2 to 8. The quadrant volumes were used to quantitatively define the symmetry with specific ratios such as the anterior symmetry ratio (ASR) (Q1 volume/Q2 volume if Q2 was greater than Q1 or Q2 volume/Q1 volume if Q1 was greater than Q2; %) and the posterior symmetry ratio (PSR) (Q3 volume/Q4 volume if Q4 was greater than Q3 or Q3 volume/Q4 volume if Q3 was greater than Q4; %) [4,12].

### 2.4. Study Design and Statistical Analyses

Each parameter was tested for normality using the Shapiro–Wilk normality test and each required test was performed. The differences between the groups were examined using the Mann–Whitney U test. Fisher’s exact test was used for the nominal variables to examine the background. A univariate logistic regression analysis was performed to test for continuous variables.

The parameters of the infants were measured three times at T1, T2, and T3. The progress was examined and Friedman’s test was used for the statistical analysis. Bonferroni corrections were used to examine the differences between the time points. McNemar’s chi-squared test with a continuity correction and Cochran’s Q test were used to test the proportions.

Severe cases with a CVAI > 8.75% at T3 were extracted and the parameters at T1 were compared and analyzed. A receiver operating characteristic (ROC) curve was drawn for the identified parameters with significant differences to confirm the thresholds. The ROC curve cut-off value was the point at which the distance from the point at the upper left corner was minimal. A Spearman’s rank correlation was used to determine the correlation coefficients. A logistic regression analysis was performed for the multivariate analysis. All statistical calculations were performed using EZR statistical software (64 bit, version 1.54) [18].

## 3. Results

During the study period, 184 infants visited our hospitals and underwent 3D head scanning. Figure 2 shows the flowchart of the enrolled infants. Sixteen patients with a gestational age < 37 weeks were excluded. One African-American infant was excluded from the study. None of the patients had a history of neonatal asphyxia. The 167 infants were included at T1 (mean age, 35.6 ± 6.3 days). Of these, 57 individuals dropped out of the study; therefore, 110 individuals were measured using the 3D scanner at T2 (mean age, 99.3 ± 9.9 days). Of the 110 individuals, 5 were excluded from the study because severe DP was diagnosed and CHT was started. In the next phase, an additional 13 individuals dropped out; therefore, 92 underwent 3D scanning at T3 (mean age, 188.5 ± 11.1 days). Only 3 of the 92 infants started CHT after instrumentation; therefore, they were not excluded from this study. Of the 167 infants initially considered, 8 (4.8%) started CHT.

### 3.1. Background Characteristics

The background characteristics of the infants are presented in Table 1. None of the children had any abnormalities during the one month medical examination and none had muscular torticollis. No developmental abnormalities were observed at T3. The cervical fixation was complete.

Parental mindfulness was not associated with DP-related morbidity (odds ratio, 0.61; 95% confidence interval, 0.13–2.60; *p* = 0.52).

### 3.2. Changes in Symmetry-Related Cranial Parameters

The Shapiro–Wilk normality test revealed that all parameters were nonparametrically distributed. The median values of these parameters are presented in Table 2. As there were no sex differences in these parameters, the values of sex were omitted.

The mean ages at the time when the measurements were performed for the 92 infants were 37.8 ± 6.4 days, 99.2 ± 8.6 days, and 188.5 ± 11.1 days at T1, T2, and T3, respectively. According to the Friedman’s test for progress, the significance of all progress was recognized for all parameters (*p* < 0.01). We observed a significant worsening of CA and the PSR from T1 to T2 during comparisons using a Bonferroni adjustment. From T2 to T3, significant improvements in the values were observed for all parameters. There were significant improvements in the ASR from T1 to T3. No significant differences in CA, the CVAI, and the PSR were observed between T1 and T3.

There was no significant association between background factors of interest and DP incidence at T2.

### 3.3. Prevalence and Severity of DP

The prevalence rates of DP were 50.0%, 56.5%, and 44.6% at T1, T2, and T3, respectively (*p* = 0.034, Cochran’s Q test; Table 2). From T1 to T2, no significant differences were observed (*p* = 0.63). From T2 to T3, a significant improvement was observed (*p* = 0.029); however, no significant differences were observed between T1 and T3 (*p* = 1.0).

The contingency tables for DP severity at T1 and T3 are shown in Table 3. The severity was classified based on a CVAI cut-off of 6.25%; 68 (73.9%) of the 92 infants had the same DP severity at T1 and T3. McNemar’s chi-squared test with a continuity correction indicated that *p* = 1.0, which also revealed no significant changes in the severity at T1 and T3. The sensitivity of a CVAI > 6.25% at T1 indicating moderate or more severe DP (CVAI > 6.25%) at T3 was 0.61; the specificity was 0.80, the positive predictive value was 0.61, and the negative predictive value was 0.80 (odds ratio, 6.3; 95% confidence interval, 2.24–18.9; *p* < 0.001).

Eight infants (8.7%) had severe or very severe DP (CVAI > 8.75%) at T3 (severe, *n* = 5; very severe, *n* = 3).

### 3.4. Predictive Threshold of Severity at T1

The measurements of eight patients in the target group who had severe or other deformities (CVAI > 8.75%) at T3 were compared with their measurements at T1 (Table 4). We observed significant differences in CA, the CVAI, and the PSR. The ROC curves for each significant difference are shown in Figure 3. The outcome was a CVAI > 8.75% at T3 and the variable used for the prediction was the T1 value of each parameter. The thresholds at T1 were 10.2 mm, 7.76%, and 87.9% for CA, the CVAI, and the PSR, respectively. The progress diagrams of the parameters with the thresholds of CA 10 mm, CVAI 7.8%, and PSR 88% at T1 are shown in Figure 4. Patients who were above the threshold at T1 worsened temporarily at T2 but many improved at T3; those who had mild DP at T1 had mild DP at T3.

A multivariate analysis of the patients with moderate or more severe DP at T3 as the objective variable and a CVAI > 7.8% at T1 as well as a PSR < 88% as explanatory variables was performed. CA was affected by the physique, and CA and the CVAI were strongly correlated (r = 0.993; *p* < 0.01); therefore, CA was excluded from this analysis. A risk was observed with an odds ratio of 3.88 (95% confidence interval, 1.06–14.3; *p* = 0.04) for a CVAI > 7.8% and an odds ratio of 6.27 (95% confidence interval, 2.11–18.6; *p* < 0.001) for a PSR < 88%.

## 4. Discussion

A 3D scanner was used to measure the natural course of cranial geometry in 92 healthy infants aged 1–6 months. It was found that a few deformation index parameters temporarily worsened from one to three months and that all parameters improved from three to six months. The incidence and severity of DP were not significantly different at one and six months. Additionally, the prediction threshold at one month of age for infants with severe DP was proven (CVAI > 7.8% at T1, PSR < 88%). To the best of our knowledge, this study is the first to longitudinally measure the course of cranial deformation parameters in healthy infants in Japan during the early age of life using a 3D scanner.

### 4.1. Characteristics of the Infants

Factors such as primiparity, multiple pregnancies, delivery method, gestational diabetes, fetal presentation, infant sex, and preterm birth have been reported to increase the risk of DP during the first four months of life [19,20,21,22,23,24]. During this study in Japan, no correlation was found between these factors, each parameter, or the prevalence of DP at T2. The current study included infants born after 37 weeks of gestation. Therefore, risk factors such as severe hypertensive disorders during pregnancy, gestational diabetes, multiple births, and preterm birth before 37 weeks of gestation were not adequately investigated. The results of this study indicated risk factors for mild to moderate disease in term infants; therefore, associations between maternal risk factors and DP should be clarified by future studies by including preterm infants.

In this study, no sex differences were observed in all parameters. There are various reports about sex differences in DP. A systematic review by De Bock et al. reported that males are at a higher risk for DP [25]. In another study using CT by Foster et al., no sex difference in the CVAI was reported and there was a significantly higher CVAI in Asian races compared with other races in children under 24 months of age [26]. In a further study, it was considered necessary to conduct a large-scale survey that included racial and sex differences.

No significant concern was noted in the parents about the cranial shape of their infant. It was presumed that the concerns of parents would not prevent the development or help in the improvement of DP. However, during the present study, it was considered that educational interventions regarding cranial shape were not performed and that even if the cranial shape was worrisome, the existence of such interventions was unknown. Aarnivala et al. reported that DP development could be controlled at three months of age using an intervention involving the neonatal milieu, positioning, and handling during the early postnatal period [27]. Therefore, thorough education regarding the skull shape may be important in the future.

### 4.2. Natural Course and Prediction of DP

Skull asymmetry usually develops and worsens during the first few weeks of life and the prevalence of DP has been reported to peak at two to three months of age before gradually decreasing. Likewise, the degree of asymmetry generally decreases after that age [10,27,28,29]. During this longitudinal study, we found that CA and the PSR worsened from T1 to T2. The occipital region of infants without cervical fixation during this period is the most susceptible to gravity in the supine position, which was considered to be the cause of the remarkable change in the PSR. In Japan, it was traditionally considered that the change in the ASR was less and the change in the PSR was greater because infants were often raised in the supine position [5]. However, the overall prevalence was not significantly different between T1 and T2. In Japan, effective DP prevention is expected to be implemented according to the guidelines mentioned by Aarnivala et al. [27] as well as tummy time from birth to three months of age, when CA worsening and PSR worsening are recognized [30].

From T2 to T3, both the symmetry-related parameters and the prevalence of DP improved significantly. These improvements seemed to be influenced by cervical fixation when the infants were able to independently change their orientation.

During comparisons with the values at T1, the CA and PSR values were similar at T3 (CA: 6.4–6.8 mm, *p* = 0.76; PSR: 92.8%–91.4%, *p* = 1.00). It was suggested that the skull deformed during T1 and T2 and recovered to its original T1 condition during T2 or T3. Only the ASR values improved over time. This seemed to be a manifestation of the characteristics of Japanese parenting.

Based on the CVAI, the severity at T3 was not significantly different from that at T1 and it did not increase in patients with a mild deformation at T1. At T3, 73.9% of the infants had the same DP severity as that at T1. Furthermore, the specificity, positive predictive value, and negative predictive value showed good results. When the same calculation was used for the PSR, it was revealed that a PSR < 88% at T1 had a good sensitivity (0.71), specificity (0.80), positive predictive value (0.65), and negative predictive value (0.85) for moderate or more severe DP at T3. The severity of DP at T3 could be predicted using the CVAI and PSR levels at T1. This seemed to aid in determining when to introduce CHT.

### 4.3. Initiation of CHT

Previous studies performed in other countries [10,27,28,29] proved that DP worsened closer to the age of three months; this was also found to be true in Japan. Similarly, there was a trend towards improvement thereafter as the natural course of development progressed. The initiation of CHT for DP is recommended before the infant reaches six months of age when the cranial sutures are not closed [4,5,6,7,8,9]. Whether the worst value of each symmetry-related parameter at three months of age can be used as a guide for starting treatment is controversial. Therefore, CHT may be delayed after several observations of natural history [31]. Noto et al. studied patients with severe DP (CA > 12 mm) and reported that 66% of children did not experience an improvement through the natural course of development alone, and that CHT resulted in a three-fold improvement in comparison with the baseline values [6]. Therefore, infants with severe deformities should be prioritized for treatment. In this study, we found that the severity at T3 was not significantly different from that at T1. Although it was thought that understanding the skull geometry from an early age of infancy could lead to the prediction of the future cranial geometry, the use of 3D scanners for medical reasons has not been approved in Japan; therefore, it was not possible to perform these examinations. Research on this subject must become more popular before 3D scanners can be approved for use. It has been considered that a convenient cranial shape-gauging tool made readily available to families and 3D measurement values provided through medical examinations would be helpful when making decisions about CHT implementation.

During this study, CHT was initiated in 4.8% of the patients. This number should not be ignored during a follow-up study of “normal” newborns. As mentioned above, the supine child-rearing position has been common in Japan for a long time and, as noted by Takamatsu et al., there is probably a foundation of cranial deformity not only racially but also in terms of lifestyle [5]. In Japan, the number of facilities where CHTs are currently available is limited; however, as education regarding cranial deformation becomes more widespread, the number of patients admitted to medical facilities will increase. Parents and healthcare professionals should be aware of the need for the treatment of cranial deformities in infants.

The optimal timing of CHT initiation based on the severity of the condition must be further examined. A larger study aimed at reducing the prevalence of DP is also necessary, which should include educational interventions for parents at the time of childbirth. The importance of having a clear record of the shape of the skull from an early stage was clarified in this study and the development of a simple tool accessible to families to understand skull shape is necessary.

### 4.4. Limitations

This study had certain limitations such as measurement accuracy. Another limitation was the use of 3D scanners, which are not yet advocated to be used for medical purposes in Japan. During this study, the measurements were obtained from each participant only once per individual. Comparative studies on the accuracy of facilities and measurements are too few. Furthermore, it is unlikely that measurement errors caused by infant movement can be eliminated. Although the Artec Eva is a handheld scanner, it has been reported to be equally accurate [32] as a nonportable 3D scanner and clinically viable for facial scanning [33]. The effects of movement were minimized by the binding of the overlapping parts. This 3D scanner, which is used to evaluate cranial deformities [34], was well-tolerated in this study. During this study, we did not provide parents with educational interventions regarding the cranial shape of infants. Furthermore, confounding factors such as the posture of the infant, position during breastfeeding, whether parents were concerned about the cranial shape of the infants, and whether the parents voluntarily performed positioning were not adequately examined.

The education of parents on the correction of the cranial shape was not considered in this study. In Japan, sufficient knowledge of DP is not yet widespread among parents and primary care providers. The failure to educate parents about DP at the start of this study was not an intervention but the norm. The cranial shape was fully explained when obtaining the informed consent before the study, suggesting that the target parents were more attentive to the cranial shape than Japanese parents in general.

In the present study, we focused on the CVAI and only examined DP; however, it is necessary to examine the course of the cephalic index and study brachycephaly in the future. In addition, the exclusion of cases in which CRT was initiated at T2 was considered to be a limitation when studying critical infants.

Although the required sample size criteria for the various statistics were met, the overall sample size was small and it is necessary to conduct a large-scale study in the future.

## 5. Conclusions

A 3D scanner was used to measure the natural course of cranial geometry changes in healthy infants aged 1–6 months. Deformation-related parameters and DP morbidity tended to improve with a peak occurring at three months. The cranial shape at one month of age was used to predict the cranial shape at six months of age. Infants with mild DP were not prone to severe DP development. Early postnatal 3D scanning of the cranial shape with properly defined cut-off values can predict the severity of deformational plagiocephaly at a later time.

## Figures and Tables

**Figure 1 jcm-11-01797-f001:**
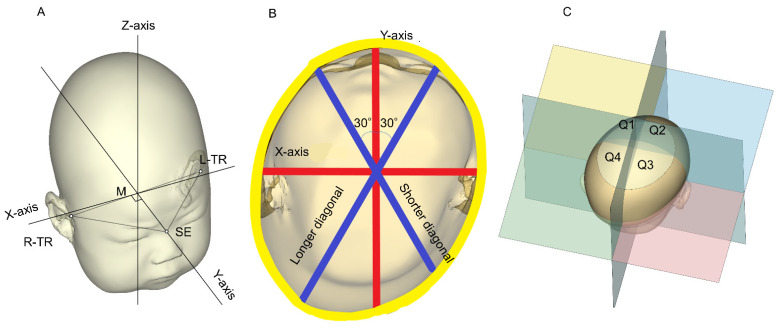
Three-dimensional images. This figure is a modified version of the figure obtained from [6]. (**A**) Methods used to determine the reference plane (level 0), X-axis, Y-axis, and Z-axis. (**B**) Cross-sectional view of level 3. Methods used to determine cranial asymmetry are shown. (**C**) Methods used to determine the anterior symmetry ratio and posterior symmetry ratio. M: midpoint between the tragions; L-TR: left tragion; R-TR: right tragion; SE: sellion; TR: tragion; Q: quadrant.

**Figure 2 jcm-11-01797-f002:**
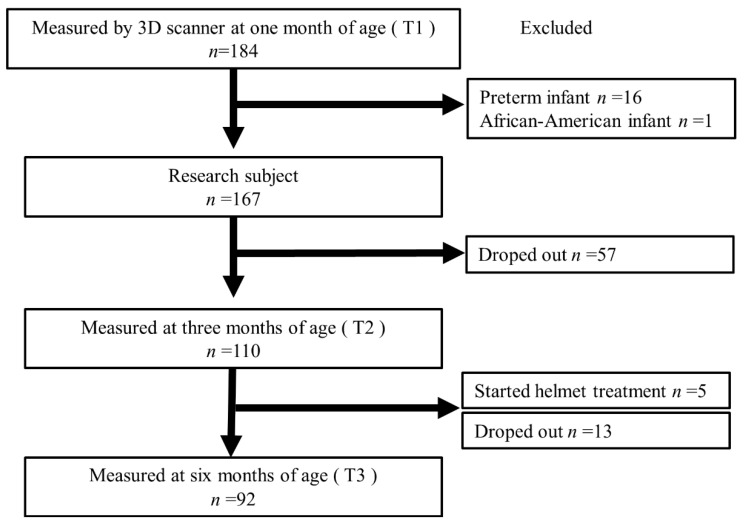
Flowchart of the enrolled infants. CHT: cranial helmet therapy; T1: first measurements at 1 month; T2: second measurements at 3 months; T3: third measurements at 6 months.

**Figure 3 jcm-11-01797-f003:**
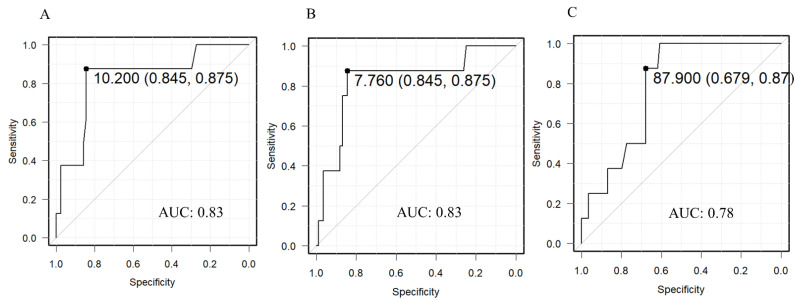
Receiver operating characteristic curves. (**A**) Cranial asymmetry. (**B**) Cranial vault asymmetry index. (**C**) Posterior symmetry ratios. AUC: area under the curve.

**Figure 4 jcm-11-01797-f004:**
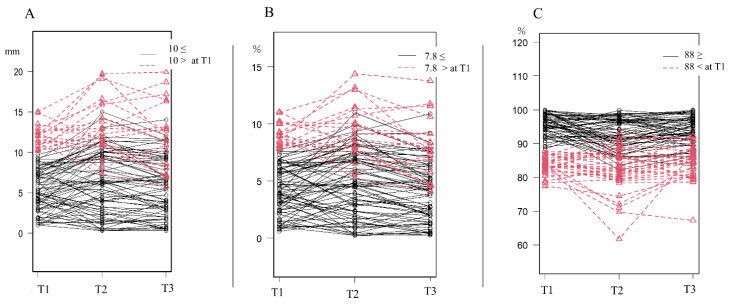
Progress diagram of each parameter with the thresholds used at T1. (**A**) Cranial asymmetry. (**B**) Cranial vault asymmetry index. (**C**) Posterior symmetry ratios. T1: first measurements at 1 month; T2: second measurements at 3 months; T3: third measurements at 6 months.

**Table 1 jcm-11-01797-t001:** Clinical characteristics of the infants (*n* = 92).

Maternal Age, Years	34.0 ± 5.3
Maternal BMI, kg/m^2^	21.8 ± 3.5
First birth/previous birth	44 (47.8)/48 (52.2)
Single pregnancy/multiple pregnancies	86 (93.5)/6 (6.5)
Infertility treatment	13 (14.1)
Hypertensive disorders of pregnancy	8 (8.7)
Gestational diabetes mellitus	13 (14.1)
Smoking history	17 (18.5)
Delivery method:	
Vaginal	46 (50.0)
Cesarean	36 (39.1)
Vacuum or forceps	10 (10.9)
Cephalic fetal presentation/other fetal presentation	85 (92.4)/7 (7.6)
Sex, male/female	48 (52.2)/44 (47.8)
Gestational age, weeks	38.5 ± 1.3
Birth weight, g	3016 ± 324
5 min Apgar score	9 ± 0.4
Breastfeeding only/formula mix	20 (21.7)/72 (78.3)
Not concerned/concerned	81 (88.0)/11 (12.0)

For each parameter, the mean ± standard deviation or number (percentage) is presented. BMI: body mass index.

**Table 2 jcm-11-01797-t002:** Changes in symmetry-related cranial parameters.

	T1	T2	T3	*p*-Value		
				T1–T2	T2–T3	T1–T3
Cranial asymmetry, mm	6.4 (4.0–9.0)	8.0 (4.1–11.3)	6.8 (3.5–10.1)	<0.01	<0.01	0.76
Cranial vault asymmetry index, %	5.0 (3.1–7.0)	5.8 (3.0–8.0)	4.7 (2.5–7.1)	0.10	<0.01	0.56
Anterior symmetry ratio, %	93.8 (89.5–96.7)	94.2 (90.8–97.1)	95.9 (92.5–98.0)	0.13	<0.01	<0.01
Posterior symmetry ratio, %	92.8 (84.6–95.5)	90.2 (84.4–96.1)	91.4 (86.2–95.5)	<0.01	<0.01	1.00
Prevalence of deformational plagiocephaly, %	50.0	56.5	44.6	0.63	0.03	1.00

All parameters are shown as median values (interquartile range) and the prevalence of deformational plagiocephaly is shown as a ratio. Comparisons between each time point were performed using the Bonferroni adjustment in Friedman’s test for progress. The prevalence of deformational plagiocephaly was calculated using a Bonferroni adjustment in Cochran’s Q test and McNemar’s chi-squared test. T1: first measurements at 1 month; T2: second measurements at 3 months; T3: third measurements at 6 months.

**Table 3 jcm-11-01797-t003:** The severity at T1 and T3.

		T3	
		CVAI ≤ 6.25%	CVAI > 6.25%
T1	CVAI ≤ 6.25%	49 (53.2%)	12 (13.0%)
	CVAI > 6.25%	12 (13.0%)	19 (20.7%)

McNemar’s chi-squared test *p* = 1.0; CVAI: cranial vault asymmetry index; T1: first measurements at 1 month; T3: third measurements at 6 months.

**Table 4 jcm-11-01797-t004:** Comparative study of the severe group.

		CVAI ≤ 8.75% at T3	CVAI > 8.75% at T3	*p*-Value
	*n*	84	8	
T1	Cranial asymmetry, mm	6.0 (3.8–8.3)	10.4 (10.2–12.7)	<0.01
	Cranial vault asymmetry index, %	4.7 (3.0–6.6)	8.3 (7.9–9.5)	<0.01
	Anterior symmetry ratio, %	93.9 (89.6–97.0)	90.3 (88.8–92.1)	0.19
	Posterior symmetry ratio, %	94.1 (84.9–97.1)	86.1 (82.7–87.8)	<0.01

For each parameter, the median value (interquartile range) is presented. CVAI: cranial vault asymmetry index; T1: first measurements at 1 month; T3: third measurements at 6 months.

## Data Availability

Not applicable.

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
