# Peer review of "Cranial Shape in Infants Aged One Month Can Predict the Severity of Deformational Plagiocephaly at the Age of Six Months"

_jcm, 2022, doi:10.3390/jcm11071797_

Round 1
Reviewer 1 Report
- brief summary The aim of the paper is to evaluate the risk of plagiocephaly at the age of 6 months based in the cranial shape at the age of 1 month.
-
General concept comments
Article: The article is very easy to read - clear language. The study desgin is well presented as the methods used. The techonological resources employed were well described and the statisitcal analysis chose were appropriated.
Review: You were very clear descrbing the exclusion criteria and they were very important. Maybe in your discussion or conclusion you should include some comments about ethnical features.
- Specific comments : the same above
Author Response
Response to Reviewer #1:
We would like to thank the reviewer for the thoughtful review and informative comments.
You were very clear describing the exclusion criteria and they were very important. Maybe in your discussion or conclusion you should include some comments about ethnical features.
Response: Lines 321-324: Added background on Japanese child-rearing, racial factors, and lifestyle.
“As mentioned above, the supine child-rearing position has been common in Japan for a long time, and as expected by Takamatsu et al., there was probably a foundation of cranial deformity, not only racially but also in terms of the lifestyle [5].”
Again, we thank you for your useful comments on our paper. We hope that the revised manuscript is suitable for publication.

Reviewer 2 Report
- Please provide the actual technical procedure of 3D scanning and data acquisition in detail.
- Please provide the detailed process of data analysis such as software operating with data.
- Please provide clinical case presentation.
- There is no suggested cut-off value, if possible provide cut-off value and follow-up strategy.
Author Response
Response to Reviewer #2:
We thank the reviewer for their thoughtful review and insightful comments.
1: Please provide the actual technical procedure of 3D scanning and data acquisition in detail.
Response: The fact that the child is held by the mother and data is obtained from the surroundings using a hand-held scanner is described on lines 75–79. The fact that the STL of the entire image is created by matching common areas is also described in lines 83–85.
In practice, as shown in the figure, the scan is taken from a distance of about 50-100 cm after the cap is put on, using a handheld 3D. During this time, the child is often resting in the mother's arms. Although we occasionally encounter a child who chirps, we continue the scan and create a complete picture by matching common areas, as described above.
2: Please provide the detailed process of data analysis such as software operating with data.
Response: Data analysis software from STL files is currently an outside secret of Japan Medical Company. We are currently in the process of approaching them about open sourcing. The analysis method within the outlined software is as described in the paper.
When the above STL file is run through JMC's analysis software, the following report is generated.
The description of analysis software made by JMC was added to line88.
“Data were analyzed using the Artec Studio image analysis software (Artec Inc., Luxembourg, Luxembourg) and original analysis software by Japan medical company(Japan Medical Company, Tokyo, Japan)to obtain 3D images and determine the cranial shape.”
3: Please provide clinical case presentation.
Response: A study of actual treatment cases is reported by co-author Noto.
Reference 6: Noto T, Nagano N, Kato R, Hashimoto S, Saito K, Miyabayashi H, Sasano M, Sumi K, Yoshino A, Morioka I. Natural-course evaluation of infants with positional severe plagiocephaly using a three-dimensional scanner in Japan: comparison with those who received cranial helmet therapy. J. Clin. Med. 2021, 10 (16), 3531. https://doi.org/10.3390/jcm10163531
4: There is no suggested cut-off value, if possible provide cut-off value and follow-up strategy
Response: Criteria for DP severity are listed on lines 108–110.
The results of this study, the threshold at T1, CVAI 7.8%, and PSR 88%, are newly highlighted on lines 242.
“Additionally, the prediction threshold at one month of age for infants with severe DP was proven (CVAI > 7.8% at T1, PSR < 88%).”
Since this study involves following up on normal newborns, there is no further follow-up. However, those indicated for helmet therapy were followed up and examined using a 3D scanner every month. In the future, we are considering reevaluating children who exceeded the cut-off value at 1 to 2 months of age at 1 month and considering CHT if there is a poor improvement.
Again, we would like to thank you for your valuable comments on our paper. We hope that the revised manuscript is suitable for publication.

Reviewer 3 Report
The paper entitled "Cranial shape in infants aged one month can predict the severity of deformational plagiocephaly at the age of 6 months" is very well written, clearly presented, and impactful for the scientific community. In this manuscript, the authors need to analyze more literature to discuss differences in CVAI in DP patients between males and females in the "discussion section".
Minor: Authors need to improve table and figure legend for Table#3 and Figure #3 accordingly.
Author Response
Response to Reviewer #3:
We thank the reviewer for the thoughtful review and your comments.
1: In this manuscript, the authors need to analyze more literature to discuss differences in CVAI in DP patients between males and females in the "discussion section".
Response: Thank you for your suggestion. Regarding sex differences, we have added references and added the following Line256-262.
“In this study, no sex differences were observed in all parameters. There are various reports about sex differences in DP. A systematic review by De Bock et al. reported that males are at higher risk for DP [25]. In another study using CT by Foster et al. reported that no sex difference in CVAI and significantly higher CVAI in Asian races compared to other races in children under 24 months of age [26]. In the further study, it was considered necessary to conduct a large-scale survey, including racial and sex differences.”
2: Authors need to improve table and figure legend for Table#3 and Figure #3 accordingly.
Response: Thank you for your suggestion. We have made the following corrections.
Table3
McNemar’s chi-square test p = 1.0; CVAI, cranial vault asymmetry index; T1, measurement of the cranial shape at 1 month of age; T3, measurement of the cranial shape at 6 months of age
Figure3
Receiver-operating characteristic curve. (A) Cranial asymmetry. (B) Cranial vault asymmetry index. (C) Posterior symmetry ratios. AUC, area under the curve.
Again, we thank you for your useful comments on our paper. We hope that the revised manuscript is suitable for publication.

Reviewer 4 Report
These authors have conducted a study to determine if severity of deformational plagiocephaly at 6 months could be predicted at 1 month of age.
- There was a high drop out rate between T1 and T2 and T2 and T3. Can the authors please explain this?
- What is “cervical fixation” (under Background Characteristics)?
- The authors report on “DP morbidity” in the Background Characteristics section. What is meant by this?
- The ROC curves used to determine severity thresholds are based on very few patients.
- In the discussion, the authors comment that “educational interventions regarding cranial shape were not performed.” Is this to mean that the investigators did not provide education or recommendations for treatment? Presumably, the families received education from their primary care providers. Absence of information on these interventions is a significant limitation to this study.
- By focusing on CVAI, the authors are limiting their study to deformational plagiocephaly and ignoring deformational brachycephaly. They should comment on this in the discussion.
- It is critical to differentiate between deformational plagiocephaly and craniosynostosis. While this is easily done with a clinical examination, the authors should comment on how this was done and by whom.
- 8% of patients ended up being excluded because of initiation of helmet therapy. Presumably, these were among the more severely affected infants. The ROC curves were derived from only the 8 most severely affected children who remained in the study. How does the exclusion of 5 patients affect the interpretation of the results derived from these 8 patients?
- The authors comment that helmet therapy should be initiated before 6 months of age because sutures are still open. Actually, helmet therapy is started that young to take advantage of rapid brain and skull growth. All except the metopic suture remain open well beyond 6 months of age.
Author Response
Response to Reviewer #4
We thank the reviewer for the thoughtful review and your comments.
1: There was a high drop out rate between T1 and T2 and T2 and T3. Can the authors please explain this?
Response: This study is a six-month follow-up of subjects who were enrolled between April 2020 and April 2021. During this period, a large-scale COVID19 pandemic broke out in Japan, a state of emergency was declared, and people were asked to refrain from going out. This may be the reason why the number of dropouts was very high. As we mentioned in the acknowledgments, we are extremely grateful to the subjects who, amid such a pandemic, visited the hospital three times for the study.
2: What is “cervical fixation” (under Background Characteristics)?
Response: That means "Baby could hold its head up." We had this expression evaluated by native English speakers and accordingly used this vocabulary.
3: The authors report on “DP morbidity” in the Background Characteristics section. What is meant by this?
Response: We explained that there was no correlation between the presence or absence of DP at T2, which has the highest incidence, and patient background. However, as you pointed out, we found it difficult to understand the meaning at this position and moved it to line 182.
“There was no significant association between background factors of interest and DP incidence at T2.”
4: The ROC curves used to determine severity thresholds are based on very few patients.
Response: Calculated with an AUC of 0.8, a significance level of 5%, a power of 90%, and a control: patient ratio of 10.5, the required sample size was 7.88 for patients and 82.75 for controls. The minimum sample size was met; however, the number of ROC patients should be at least 10, which, as you point out, is a small number of studies. The overall number of samples is low, which has been acknowledged as a Limitation (lines 360–362).
“Although the required sample size criteria for the various statistics were met, the overall sample size was small, and it was considered necessary to conduct a large-scale study in the future.”
5: In the discussion, the authors comment that “educational interventions regarding cranial shape were not performed.” Is this to mean that the investigators did not provide education or recommendations for treatment? Presumably, the families received education from their primary care providers. Absence of information on these interventions is a significant limitation to this study.
Response: In Japan, DP knowledge is not widespread enough in general. There is not enough dissemination of knowledge, including about helmet therapy, among primary care providers. At this time (March 2022), only three papers have been published from Japan (Refs. 4–6) on this topic. I cannot find any papers in Japanese in Japan either. So much so that there is currently a lack of awareness of DP. We are working hard to promote awareness regarding DP and helmet therapy.
As of 2020, when this study was initiated, there was no primary care for cranial geometry in Japan as a whole. When obtaining informed consent for the study, the DP was fully explained to the parents, and we did not consider it an intervention at that time. Educational intervention is considered an issue for further study and is being implemented.
In light of your suggestion, we have added this as a Limitation in the revised manuscript (lines 350–355).
“Education of parents on correction of the cranial shape was not considered in this study. In Japan, sufficient knowledge of DP is not yet widespread among parents and primary care providers. The failure to educate parents about DP at the start of this study was not an intervention but the norm. Conversely, the cranial shape was fully explained while obtaining the informed consent before the study, suggesting that the target parents were more attentive to cranial shape than the general Japanese parents.”
6: By focusing on CVAI, the authors are limiting their study to deformational plagiocephaly and ignoring deformational brachycephaly. They should comment on this in the discussion.
Response: As you pointed out, we intentionally did not consider the Cephalic index in this study. This is because the diagnostic criteria for brachycephaly in Japan differ from those in other countries, and it is necessary to consider this issue separately. The following reference indicates a study of the Japanese Cephalic index. It discusses racial differences between Japanese and Westerners. This was added to the Limitation (lines 356–358).
Koizumi T, Komuro Y, Hashizume K, Yanai A: Cephalic index of Japanese children with normal brain development. J Craniofac Surg 21: 1434–1437, 2010
“In the present study, we focused on CVAI and only examined DP; however, it was considered necessary to examine the course of the Cephalic Index and study brachycephaly in the future.”
7: It is critical to differentiate between deformational plagiocephaly and craniosynostosis. While this is easily done with a clinical examination, the authors should comment on how this was done and by whom.
Response: As you pointed out, the differentiation of craniosynostosis is crucial. In this study, a pediatrician performed examinations at the time of 3D imaging to confirm the absence of craniosynostosis. This has been added to line 71 in the Methods section.
“At each measurement, the pediatrician examined the skull sutures and confirmed the absence of craniosynostosis. Finally, Computed Tomography imaging was performed only in cases where CHT was necessary.”
8: 8% of patients ended up being excluded because of initiation of helmet therapy. Presumably, these were among the more severely affected infants. The ROC curves were derived from only the 8 most severely affected children who remained in the study. How does the exclusion of 5 patients affect the interpretation of the results derived from these 8 patients?
Response: Thank you for pointing this out. For longitudinal follow-up, we tested only the values for affected children measured at T3 within this study. The following is a comparative study of the 167 subjects at T1 with and without helmets. As noted in the text, there were eight children for whom helmets were ultimately used. The results were compared with 159 controls.
Same results as in Table 4. Furthermore, if we draw the ROC for each parameter with helmet therapy for the outcome, we obtain the following results.
The thresholds for CA, CVAI, and PSR were 10.3 mm, 6.3%, and 84.3%, respectively. These are the likely effects of excluding helmet therapy from the study.
The presence or absence of CRT is an indeterminate factor, not only because of the parameters but also because of confounding factors such as family economic circumstances. In this manuscript, we have decided to classify the children measured at T3 based on parameters with no uncertainties. This has been added to the Limitation section (lines 358–359).
“In addition, the exclusion of cases in which CRT was initiated at T2 was considered a limitation when studying critical infants.”
9: The authors comment that helmet therapy should be initiated before 6 months of age because sutures are still open. Actually, helmet therapy is started that young to take advantage of rapid brain and skull growth. All except the metopic suture remain open well beyond 6 months of age.
Response: Thank you for pointing this out. As you mentioned, we believe that in some children, correction by CHT is possible beyond 6 months of age.
Again, we would like to thank you for your valuable comments on our paper. We hope that the revised manuscript is suitable to be considered for publication.

Round 2
Reviewer 2 Report
Manuscript has been revised sincerely and adequately supported by additional descriptions. Can the suthors suggest cut off value of CVAI and timing for starting helmet therapy if it is possible?